# Toward Renewable-Based Prebiotics from Woody Biomass: Potential of Tailored Xylo-Oligosaccharides Obtained by Enzymatic Hydrolysis of Beechwood Xylan as a Prebiotic Feed Supplement for Young Broilers

**DOI:** 10.3390/ani13223511

**Published:** 2023-11-14

**Authors:** Ines Dieryck, Winnie Dejonghe, Wouter Van Hecke, Joy Delacourt, An Bautil, Christophe M. Courtin, Daniel Vermeulen, Johan Buyse, Jan Paeshuyse

**Affiliations:** 1Laboratory of Host Pathogen Interactions, Department of Biosystems, KU Leuven, 3001 Leuven, Belgium; ines.dieryck@kuleuven.be (I.D.);; 2Flemish Institute for Technological Research, 2400 Mol, Belgium; winnie.dejonghe@vito.be (W.D.); wouter.vanhecke@vito.be (W.V.H.); 3Laboratory of Food Chemistry and Biochemistry, Department of Microbial and Molecular Systems, KU Leuven, 3001 Leuven, Belgium; an.bautil@kuleuven.be (A.B.); christophe.courtin@kuleuven.be (C.M.C.); 4Laboratory of Livestock Physiology, Department of Biosystems, KU Leuven, 3001 Leuven, Belgium; daniel.vermeulen@kuleuven.be (D.V.); johan.buyse@kuleuven.be (J.B.)

**Keywords:** xylo-oligosaccharides, prebiotics, broilers, enzymatic hydrolysis, short chain fatty acid analysis

## Abstract

**Simple Summary:**

Although antibiotic resistance emerges naturally, this process has been accelerated by the worldwide overuse and misuse of antibiotics. It is essential to find effective alternatives in the broiler industry to improve poultry health while maintaining production efficiency and product safety. In this study, we aimed to evaluate a potential alternative: wood-derived component xylo-oligosaccharides. The objective of this research was to investigate the potential of xylo-oligosaccharides as a prebiotic feed supplement for broilers. Therefore, a pilot study was conducted to explore the optimal xylo-oligosaccharide profile by in vitro fermentation. Subsequently, xylo-oligosaccharides with an optimal profile were produced in large quantities, and an animal feed experiment was performed. During this experiment, the growth performance, feed conversion ratio, and intestinal parameters of the broilers were evaluated for 15 days. Results from the pilot study indicated that higher enzyme concentrations in the production process yield a product that leads to a higher butyric acid production during in vitro fermentation by caecal bacteria. Supplementation of the tailored xylo-oligosaccharides to the broiler diet resulted in higher *Bifidobacterium* counts, bacteria beneficial to the health of birds, from day 11 onwards.

**Abstract:**

Although antibiotic resistance emerges naturally, this process has been accelerated by the worldwide overuse and misuse of antibiotics. It is essential to find effective alternatives in the broiler industry to improve poultry health while maintaining production efficiency and product safety. In this study, we aimed to evaluate a potential alternative: wood-derived xylo-oligosaccharides (XOS). The objective of this research was to investigate the potential of XOS prepared using enzymatic hydrolysis of beechwood xylan as a prebiotic feed supplement for broilers. A pilot study was conducted to explore the optimal XOS fraction profile by in vitro fermentation. Subsequently, a semi-continuous enzyme membrane reactor was used, allowing for the production of tailored XOS in large quantities. Given the strong bidirectional relationship between intestinal health, nutrition, and intestinal microbiota composition in broilers, an in vivo experiment was performed to explore the potential of XOS as a prebiotic feed supplement by investigating growth performance, feed conversion ratio, caecal short and medium chain fatty acid (SCFA and MCFA) concentration, and microbiological composition of the caecal content. Results from the pilot study indicated that higher enzyme concentrations in the hydrolysis process yield a product that leads to a higher total SCFA and MCFA- and butyric acid production during in vitro fermentation by caecal bacteria. Supplementation of the tailored XOS to the broiler diet (day 1 (d1)-d8 0.13% *wt/wt* XOS, d9-d15 0.32% XOS) resulted in higher *Bifidobacterium* counts, beneficial to the health of birds, on d11 and d15.

## 1. Introduction

The development of antibacterial agents has been one of the major accomplishments of modern medicine. However, although antibiotic resistance emerges naturally, this process has been accelerated considerably by the worldwide overuse and misuse of antibiotics. Antibiotic resistance rates are reaching dangerously high levels in all parts of the world. A growing number of common bacterial diseases, such as pneumonia, tuberculosis, blood poisoning, gonorrhea, and foodborne diseases, are becoming harder, sometimes even impossible, to treat as antibiotics become less effective due to newly emerging resistance mechanisms that are spreading rapidly across the world. The World Health Organization even describes antimicrobial resistance as one of the biggest threats to global health, food security, and development today [1]. The widespread use of antibiotics for therapeutic, prophylactic, and production purposes in animal production systems contributes significantly to this problem. In the poultry industry, antibiotic growth promotors (AGP) have been extensively used to improve growth performance, feed efficiency, prevention of subclinical infection, and overall health because of their low cost and ease of use [2]. Due to the increasing concern for transmission of antibiotic-resistance genes to human and animal pathogens, the use of AGPs has been banned in the European Union since 2006; a practice that has also been adopted by several other countries such as Mexico, New Zealand, Chile, Turkey and the Republic of Korea [3,4,5]. The United States of America, Australia, Japan, and Canada implemented laws to partially ban or exclude some antibiotic-derived additives [5,6,7]. Currently, therapeutic and prophylactic use of antibiotics in animal production remains legal. However, in the European Union, antimicrobial medicinal products may not be administered for routine prophylaxis anymore under Regulation (EU) 2019/6, implemented in January 2022: “prophylactic use of antimicrobial medicinal products can only be used in exceptional cases, for the administration to an individual animal or a restricted number of animals when the risk of an infection or of an infectious disease is very high and the consequences are likely to be severe” [8]. Despite these efforts, the application of AGPs has been projected to augment due to the increased implementation of large-scale intensive farming operations in countries such as Brazil, China, India, Russia, and South Africa [9]. Per contra, low-dosage antibiotic usage can have several advantages, including reduction in pathogen load in the broiler house and thickening of the gastro-intestinal wall of the birds, which could result in increased nutrient absorption [2]. Therefore, it is essential to find effective alternatives that improve poultry health and maintain production efficiency and poultry product safety in order to reduce antibiotic use [3,10].

Since the early 1960s, global poultry meat production per capita has increased sixfold. Chicken meat, which contributes 90% of global poultry meat production, has undergone a significant transformation, evolving from a relatively scarce food product to an easily accessible, economically viable, protein-rich, and low-fat food product characterized by a favorable fatty acid profile [11,12]. Poultry meat has been the primary driving force behind the increase in total meat production: nearly 50% of the increase in total meat production over the last decade is due to the expanding poultry industry [13]. According to statistical data recorded by The Food and Agriculture Organization of The United Nations, global poultry meat production in 2020 reached 133.4 million tons, accounting for 38.95% of the total meat production [11]. A prominent strategy for optimizing economic returns and fostering sustainable intensification entails the enhancement of feed efficiency using intensive genetic selection and refined nutritional management. Feed expenses constitute a substantial portion, up to 70%, of the overall cost associated with modern broiler production and exhibit significant fluctuations from year to year, thereby complicating profit projections [14,15]. Furthermore, over 81% of the raw materials used in poultry diets, including cereal grains, cassava, soybean, pulses, rapeseed, soy oil, and soybean meal, are considered to be in direct competition with human nutritional needs [16]. Consequently, the improvement in feed efficiency in broilers is considered a crucial aspect of the sustainable intensification of poultry production. However, the rigorous selection for performance traits, such as feed conversion ratio (FCR), has led to substantial physiological adaptations within the gastro-intestinal system of the modern broiler. This has resulted in a range of issues, including alterations in size and histological characteristics of digestive organs at all stages of development, an impaired immune system, and diminished nutrient absorption in modern broilers, rendering the delicate balance between nutrients, microbiota, and intestinal health even more fragile [17]. Over the years, AGPs effectively mitigated dysbiosis in broilers. However, due to restrictions imposed on AGP utilization, dysbiosis has resurfaced as a significant challenge in modern poultry production [18].

Potential alternatives are prebiotic non-starch polysaccharides (NSP), non-digestible but fermentable feed additives that stimulate the growth of microbiota that are beneficial to the host health, provide substrates for bacterial fermentation in the ileo-caeco-colic section of the gastro-intestinal tract; inhibit colonization of pathogenic bacteria; provide energy and limiting nutrients for the intestinal mucosa; and, consequently, improve growth performance and feed efficiency of the birds [4,19,20]. Because NSPs are not degraded by the acidic gastric fluids nor by the host gastro-intestinal digestive enzymes, they reach the ileum mainly intact [21]. However, they can be hydrolyzed using carbohydrases that are available in the broiler diet [19]. Interestingly, xylo-oligosaccharides (XOS), hydrolysis products of (arabino)xylan, cannot be degraded by enteric pathogenic bacteria such as *Staphylococcus aureus*, *Clostridium difficile*, *Salmonella enterica*, and *Campylobacter jejuni* [3]. On the other hand, previous research showed that XOS could be fermented by probiotic bacterial strains such as *Bifidobacterium* spp. and *Lactobacilli* spp., generating a range of fermentation products such as short chain fatty acids (SCFA; mainly acetate, propionate and butyrate), medium chain fatty acids (MCFA) and lactate [3,21,22]. An increase in SCFA and MCFA concentration in the intestinal environment is often associated with increased fermentation; improved gastro-intestinal tract growth by regulation of growth and proliferation of epithelial cells; modulation of the host mucosal immune system, including regulation of intestinal inflammation; an increase in beneficial bacteria; and a decrease in pathogenic bacteria [19,23,24]. Moreover, SCFA butyric acid is considered an available energy source, augmenting the energy resources available for the growth of the birds [3,19,22]. In addition, butyric acid is the preferred energy source for colonocytes, stimulating the colon epithelial cells and increasing the absorptive capacity of the epithelium [25].

XOS are oligomers with a non-linear backbone containing several xylopyranosyl residues (usually 2–20) that are linked via β-(1,4)-bonds and contain different functional side groups, such as acetyl groups and uronic acids [26,27]. XOS are hydrolysis products of xylan, the second most abundant carbohydrate material after cellulose in lignocellulosic biomass [26,28,29]. Various processes for the production of XOS have been explored and include chemical, physical, enzymatic degradation, and autohydrolysis [28]. As a result of the increasing wood consumption, an abundant source of cheap lignocellulosic biomass, for example, in the form of end-of-life wood-based products, is becoming available [30]. Transforming locally available wood residues for applications for the agro-industry is an interesting approach to valorizing waste wood and requires further exploration. For this purpose, innovative conversion technologies are currently investigated, and validation of the resulting products is needed.

The objective of this research was to investigate the potential of XOS prepared by enzymatic hydrolysis of beechwood xylan as a prebiotic feed supplement for broilers. In the first part of this study, a pilot study was conducted to explore the optimal XOS fraction profile. The obtained XOS products were evaluated by in vitro fermentation. Subsequently, a semi-continuous enzyme membrane reactor (EMR) was used, allowing for the production of tailored XOS in larger quantities, and an in vivo animal experiment was performed to explore the potential of the obtained XOS product as a prebiotic feed additive by investigating growth performance, FCR, caecal SCFA and MCFA concentration, and microbiological analysis of caecal content of broiler chickens.

## 2. Materials and Methods

### 2.1. Pilot Study

#### 2.1.1. Preparation of the XOS Fractions

To select the XOS fractions that have a positive effect on the health of animals, a small-scale in vitro pilot study was performed. To produce XOS fractions with different molecular sizes, xylan derived from beechwood (BX, REF 4414.1, Carl Roth GmbH, Karlsruhe, Germany) was purchased and enzymatically hydrolyzed in a bioreactor. Briefly, a 7% *wt/wt* xylan solution in demineralized water was prepared. Enzymatic hydrolysis was initiated by adding Cellic CTec2 (Novozymes, Bagsværd, Denmark), an enzyme blend exhibiting cellulase, glucosidase, xylanase, xylosidase, and arabinosidase activity, with an optimal reaction temperature of 45 to 50 °C and optimal reaction pH of 5.0 to 5.5 [31]. Four XOS fractions (XF) were prepared by adding various enzyme amounts to the 7% xylan solution: XF139 (13.9% *vol/wt*, calculated as Cellic CTec2 (mL) per xylan (g) and corresponding to 719 U/g xylan), XF050 (0.50% *vol/wt* or 25 U/g xylan), XF005 (0.05% *vol/wt*, or 2.6 U/g xylan), and XF001 (0.01% *vol/wt* or 0.5 U/g xylan). Enzymatic hydrolysis of the samples was performed in a 3 L Applikon batch set-up (Getinge, Gothenburg, Sweden), thermoregulated at 50 °C. The acidity was kept constant at pH 5.8 by inline monitoring and adding 1 M sodium carbonate solution. The dynamic viscosity of the reaction liquid was monitored in real-time using a ReactaVisc RV3 Reaction Vessel Viscometer (Hydramotion Ltd., Malton, UK). Hydrolysis time varied for all enzyme concentrations but was approximately one day for XF139 (23.2 h), XF050 (24.1 h), XF005 (18.3 h), and XF001 (24.0 h). Xylose, xylobiose, xylotriose, xylotetraose, xylopentaose, and xylohexaose concentrations were quantified using High-Performance Anion-Exchange Chromatography with Pulsed Amperometric Detection (HPAEC-PAD). The average molecular weight of the xylan hydrolysis products was monitored using Gel Permeation Chromatography (GPC) equipped with an RID detector (Shimadzu, Tokyo, Japan) [32]. The obtained XOS fractions were stored at −20 °C until further use.

#### 2.1.2. In Vitro Fermentation of the XOS Fractions

The obtained XF were evaluated using in vitro anaerobic fermentation by caecal bacteria, followed by gas chromatography (GC) analysis of the produced fatty acids (FA). Caecal content of eight twenty-day-old male Ross 308 broiler chickens (Belgabroed NV, Merksplas, Belgium) was pooled and a 42.5% *vol/vol* mixture in sterile phosphate-buffered saline (PBS; REF D5652, Merck, Darmstadt, Germany) was prepared. Per XF preparation, 0.6 mL XF preparation, 2.4 mL sterile PBS, and 1 mL caecal content solution were added to a 50 mL tube (REF 227261, Greiner Bio-One International, Kremsmünster, Austria), resulting in a final XF concentration of 1% *wt/wt* and a final caecal content of 10.6% *vol/vol*. A blank (blank T24) was prepared by adding 3 mL sterile PBS (REF D5652, Merck) and 1 mL caecal content solution to a 50 mL tube. All tubes were incubated for 24 h at 41 °C under anaerobic conditions that were obtained by placing the tubes in an anaerobic jar equipped with an Anaerocult A system (REF 1.32381.0001, Merck). The absence of oxygen was confirmed using an Anaerotest strip (REF 1. 32371.0001, Merck) that was placed in the anaerobic jar and was regularly checked. Just before SCFA analysis of the fermentation mixtures, a blank (blank T0) with the same composition as blank T24 was prepared but not incubated at 41 °C. The fermentation mixture composition of all XF samples is summarized in Table 1. For each fermentation mixture composition, two biological replications were prepared. SCFA and MCFA analysis of all samples was performed immediately after incubation.

#### 2.1.3. SCFA and MCFA Analysis

SCFA and MCFA analysis was performed using diethyl ether extraction (DEE) of the samples, followed by GC. DEE sample preparation was based on a method described by Van Craeyveld and colleagues and was executed on ice [33]. Briefly, per XF fermentation mixture, 400 to 500 mg of the mixture and 100 µL 0.75% *vol/vol* 2-methylhexanoic acid (REF 338273, Merck) standard solution were added to a 2 mL tube. Subsequently, 200 µL 25% *vol/vol* sodium chloride (REF 3957.1, Carl Roth, Karlsruhe, Germany) solution and 200 µL 9.2 M sulfuric acid (REF 133610025, Thermo Fisher Scientific, Waltham, MA, USA) solution were added to the sample. Both steps were followed by a vigorous vortex step. After incubating for two min on ice, 800 µL diethyl ether (REF 32203, Honeywell Riedel-de Haën AG, Seelze, Germany) was added, and extraction of the SCFA was performed using alternating vortexing and ice-bath cooling the sample during three min. The tube was centrifuged for five min (2800 RCF, 4 °C), and the diethyl ether phase, containing the organic acids, was transferred to a Reacti-Vial (REF TS-13097, Thermo Fisher Scientific) containing 0.2 to 0.3 g activated anhydrous sodium sulfate (REF 204447, Merck). The vial was vortexed and centrifuged for six min (2800 RCF, 4 °C), after which the dried extract was transferred to a glass vial (SureSTART 2 mL Glass Screw Top Vial, REF 6ASV9-1P, Thermo Fisher Scientific) containing a polyspring insert (SureSTART Polyspring Insert for 2 mL Vials, REF 6EME03CPPSP, Thermo Fisher Scientific). Four standard solutions with known SCFA composition were prepared using a volatile free acid mix (REF CRM46975, Merck), and SCFA was extracted using the same protocol as described above. GC analysis was performed the same day.

SCFA and MCFA composition of the samples was determined with an HP 6890 Series GC System gas chromatograph (Agilent Technologies, Santa Clara, CA, USA). The instrument was equipped with an automatic liquid sampler (Agilent Technologies 7683 Series Injector, Agilent Technologies) for cool on-column injection, a flame ionization detector, and a capillary DB-FFAP column (Agilent J&W GC Columns, 30 m × 0.32 mm, film thickness 0.25 µm; REF HEWL123-3232, Avantor, Radnor Township, PA, USA). The sample injection volume was set at 0.5 µL. Nitrogen was used as carrier gas at a flow rate of 25 mL/min. The column temperature and the temperatures of the injector and detector were set at 55 and 245 °C, respectively. All measurements were performed in duplicate.

### 2.2. In Vivo Study

In order to assess the prebiotic properties of the produced XOS fraction, an in vivo animal feed experiment was designed.

#### 2.2.1. XOS Production 

To obtain larger quantities of an XOS fraction intended as a chicken feed supplement (XFCF) with a specific MS profile based on the results of the in vitro pilot study, the enzymatic hydrolysis of the beechwood xylan (BX) was performed in an EMR running in a semi-continuous mode as described by Ríos-Ríos and colleagues [32]. In brief, 2.6 L of BX (7% *wt/wt*) was hydrolyzed with Cellic CTec2 (5.4 U/g xylan) at batch mode during 3 h at the same reaction conditions as in the batch set-up described above: 50 °C, pH 5.8 and mixing at 300 rpm. Once the viscosity reached 2 cP (monitored in real-time using a ReactaVisc RV3 Reaction Vessel Viscometer (Hydramotion Ltd., Malton, UK)), the filtration started automatically. A membrane module (Romicon HF 1018-1.0-43-PM5) with a total membrane exchange surface of 0.09 m^2^ was used. After every cycle of 3 h batch residence time, 1.6 L of BX hydrolysate was filtered off for 140 min, after which 1.6 L of fresh BX (7% *wt/wt*) solution was added to the reactor. The reactor was run for a total time of 19 h, during which three batches of fresh substrate BX were added. Samples from the reactor and permeates were collected in the function of time and analyzed using GPC for the size of the obtained xylan fragments and by HPAEC-PAD to determine the concentration of xylose to xylohexaose. The obtained XFCF was lyophilized (DELTA 1-24 LSC, Christ, Osterode am Harz, Germany) and kept at −20 °C until further use.

#### 2.2.2. Preparation of the Experimental Diets

For this study, a commercial starter diet (18% CP) (Chicken Start mash, REF L/1.26112019.8417, AVEVE NV (Arvesta), Leuven, Belgium) with a nutrient profile that meets the requirements described in the National Research Council recommendations, was acquired [34]. The composition of the starter feed is reported in Table A1 (Appendix A). 

Three experimental diets were prepared, differing only in the supplementation of XFCF (0.0% *wt/wt* (FCON), 0.2% *wt/wt* (FXFCF0.2), and 0.5% *wt/wt* (FXFCF0.5)), and resulting in diets with final XOS concentrations of 0.0%, 0.13%, and 0.32%, respectively. In order to obtain a homogeneous feed mixture for the supplemented diets, premixtures were made by manually mixing the corresponding amount of XFCF with 20 g starter feed for three min. The premixtures were then gradually added to the final batch of feed in a concrete mixer (Euro-Mix 125, Altrad Lescha, Burgau, Germany) and mixed for fifteen min. 

#### 2.2.3. Analytical Methods for Characterization of the Feed Mixtures

In order to obtain more insight into the chemical characteristics of the feed mixtures, the following parameters were determined: DM, total monosaccharides (MS), water-extractable (WE) MS, free MS, reducing end MS, the total amount of arabinoxylan (TOT-AX), WE arabinoxylan (WE-AX) and average degree of polymerization (avDP) of the AX chain. Samples were collected from the three experimental diets and the pure XFCF preparation. DM of the samples was evaluated by calculating the feed moisture content after 15 h of drying in a hot air oven at 130 °C. 

TOT-AX and WE-AX content and composition of feed were calculated based on the L-arabinose and D-xylose profile of the samples, determined using gas chromatography according to the methods previously described by Bautil and colleagues [35]. The MS included in this study were L-arabinose, D-galactose, D-glucose, D-mannose, and D-xylose. Briefly, to measure WE-AX content in the samples, microbial enzymes present in the feed, in particular endoxylanases, were heat-inactivated. Aqueous extracts of the sample were made using a KCl-HCl buffer, and the extracts were hydrolyzed to yield monosaccharides using a trifluoroacetic acid solution. Subsequently, a reduction step with sodium borohydride and an acetylation step with acetic anhydride were performed. The resultant alditol acetates were separated using a 6890N GC System Gas Chromatograph (Agilent Technologies). The instrument was equipped with a Supelco SP-2380 polar column (30 m × 0.32 mm, film thickness 0.20 µm; REF HEWL123-3232, Supelco, Bellefonte, PA, USA), an autosampler, splitter injection port (split ratio 1:20), and a flame ionization detector. Helium was used as carrier gas. The column temperature and the temperature of the injector and detector were set at 225 and 270 °C, respectively. For TOT-AX, the procedure was similar, except for the extraction step, which was performed on the feed sample directly. All measurements were performed in duplicate. 

TOT-AX (%) was calculated as 0.88 × [(L-arabinose %) + (D-xylose %)]. Similarly, the total amount of WE-AX (%) was calculated as 0.88 × [(WE-L-arabinose %) + (WE-D-xylose %)]. A conversion factor of 0.88 was used to account for the release of water molecules by MS during the polymerization reaction. The purity of the XOS fraction was calculated as 0.88 × [D-xylose%] available in this fraction. 

Total and WE arabinose to xylose (A/X) ratios (TOT-AX A/X ratio, WE-AX A/X ratio) were calculated by dividing the arabinose by the xylose content of the samples and their aqueous extracts, respectively. These A/X ratios give an indication of the complexity of the AX chain and, hence, the likelihood of the AX polymer being hydrolyzed by the microbial AX degrading enzymes and subsequently fermented by microbiota [35]. The avDP was calculated as [(L-arabinose %) + (D-xylose %)] divided by the amount of reducing end D-xylose (%).

#### 2.2.4. Animal Management and Sample Collection

The number of animals used in this study was calculated based on data obtained from a similar experiment conducted by Courtin and colleagues [23], using the software program G*Power (G*Power version 3.1.9.7, Heinrich-Heine-Universität, Düsseldorf, Germany) according to the instructions of Faul and colleagues [36]. The following input values were used for the two-tailed power analysis test: an effect size of 1.16, a level of significance of 0.05, and a power of 0.85. The minimal number of animals per treatment per sampling moment was determined to be fifteen. An overview of the experimental design is presented in Table 2. 

A total of 96 one-day-old male Ross 308 broiler chickens (Belgabroed NV) from the same broiler breeder flock with an age of 47 weeks were purchased and housed at the KU Leuven pilot farm TRANSfarm (Lovenjoel, Belgium) under the standard housing conditions for broilers determined by the Flemish Government [37]. The ambient room temperature was kept at 34 °C until day 3 (d3), after which it was gradually decreased by 1 °C every two days for the duration of the experiment. During the first five days, a light schedule providing 23 h light/1h dark (23L:1D) was set; thereafter, a 16L:4D:2L:2D light cycle was maintained until the end of the experiment (d15). Four-floor pens, equal in size (1.5 m^2^) and separated using metal fences, were covered with wood shavings as litter. Feed and drinking water were provided ad libitum. 

Upon arrival at the broiler house, the chickens were individually tagged and randomly divided into four groups (24 chicks per group): two groups that received XOS-supplemented feed (XA and XB) and two groups that received the untreated control feed (CA and CB). Groups XA and XB received 0.2% XFCF-supplemented feed during the first eight days and 0.5% XFCF-supplemented feed from d9 onwards. Two repetitions per treatment were introduced in the experiment in order to reduce the pen effect on the obtained data. On d1, d8, d11, and d15, all individual broilers and feed leftovers per pen were weighed to calculate body weight (BW), estimated feed intake (FI, *n* = 2), and estimated FCR. Animal samples were collected on d8, d11 and d15. On each sampling day, fifteen chicks per treatment were selected at random and euthanized by decapitation. The animals were dissected to collect duodenum, jejunum, ileum, and both caeca. Duodenum, jejunum, and ileum were emptied, and their net weight was registered. The content of both caeca was pooled and divided over two 2 mL tubes: one for quantification of SCFA and one for microbiological analysis. To the latter, a 50% glycerol (REF 3783.4, Carl Roth, Karlsruhe, Germany) solution was added to obtain a final concentration of 15% *vol/vol* glycerol. Samples were stored at −80 °C until further use. 

#### 2.2.5. Microbiological Analysis of Caecal Content

Microbiological analysis of the caecal content was evaluated using selective agar plates. Three groups of caecal bacteria were enumerated: *Enterobacteriaceae*, *Bifidobacteria,* and aerobic and aerotolerant anaerobic *lactobacilli*. 

To 2 mL microtubes with screw cap (REF 72.694, Sarstedt AG and Co. KG, Nümbrecht, Germany), five to seven 2.0 mm Zirconia beads (REF 11079124zx, BioSpec Products, Bartlesville, OK, USA) per tube were added, and the entities were autoclaved. Under sterile conditions, 300–500 mg caecal content was added to the sterile microtubes, and the samples were placed on ice. Ice-cold, sterile peptone physiological salt solution (PPS; 1 g/L Bacto Peptone (REF 211677, Thermo Fisher Scientific, Waltham, MA, USA), 8.5 g/L sodium chloride (REF 3957.1, Carl Roth)) was added to obtain a concentration of 500 mg caecal content per mL PPS. In order to ensure the selective quality of the selective agar plates, bacterial strains functioning as positive and negative controls were included in the study. These controls were also prepared in sterile microtubes by adding scrapings from glycerol stocks (−80 °C) to 1 mL sterile PPS. All samples were homogenized (Precellys Evolution, Bertin Instruments) for ten sec at 4500 RPM. In 96-deep well plates (Nunc 96 DeepWell Polystyrene Plates, REF 27860696, Thermo Fisher Scientific), nine tenfold dilutions of each sample in sterile PPS were prepared, resulting in test concentrations ranging from 500 to 500 × 10^−8^ mg caecal content per mL. 

Plating of the samples was performed based on a modified version of the drop plate method as described by Chen and colleagues [38]. Briefly, selective agar plates were prepared in square Petri dishes (REF 391-0465, VWR, Radnor, PA, USA). The composition of the selective agar plates and the applied incubation conditions for each bacterial group are summarized in Table 3. 

Using a multichannel pipette, 5 µL drops of the samples were placed in a ten-by-ten grid on the plates. A schematic overview of the organization of a plate is illustrated in Figure 1. Per plate, ten samples could be tested, each in nine dilutions, and three positive bacterial controls (PC), three negative bacterial controls (NC), and four PPS sterility controls (SC) were included for plate validation. All samples were tested in triplicate on separate plates. Plates were placed in a humidified incubator at 37 °C according to the conditions described in Table 3. For the enumeration of *Bifidobacteria*, anaerobic conditions were obtained by placing the plates in an anaerobic jar equipped with the Anaerocult A system (REF 1.32381.0001, Merck). The absence of oxygen was confirmed using an Anaerotest strip (REF 1. 32371.0001, Merck) that was placed in the anaerobic jar and was regularly checked.

Following the incubation step, plates were validated on an individual basis based on three conditions: (1) growth could be observed for all PCs; (2) no growth could be detected for the NCs; and (3) no growth could be detected for the SCs. Per sample, colonies in the most adequate dilution were counted, and log10 CFU/g caecal content for each sample was determined. All measurements were performed in triplicate, and the position of each sample on the plate was randomized in order to minimize potential plate position effects. 

#### 2.2.6. SCFA Analysis

Short-chain fatty acid analysis was performed using DEE of the samples, followed by GC. DEE sample preparation and SCFA analysis of the caecal content were identical to the method described for the fermentation products in the pilot study. All measurements were performed in duplicate.

### 2.3. Statistical Analysis 

All acquired data were analyzed using the statistical software package JMP (JMP Pro version 16.0.0, SAS Institute, Cary, NC, USA). Previous to the analysis, data were screened for outliers by the ‘explore outliers’ function implemented in JMP. The normality of the data was assessed by performing a Levene test. In the case of normal distribution, homoscedasticity was tested using Barlett’s test. Data among groups were analyzed using the Student’s *t*-test in case of equal variances and the Welch’s *t*-test in case of unequal variances. In case the data were not normally distributed, the non-parametric unpaired two-samples Wilcoxon test was used. For the in vivo experiment, two repetitions per treatment were introduced in the experiment in order to reduce the pen effect on the obtained data. Before statistical analysis of the data between treatments, repetitions of the same treatment were analyzed for pen effects. In the data sets of this experiment, no pen effects were detected, and subsequently, data sets were combined per treatment before analysis. For all performed tests, a *p*-value of ≤0.05 was considered statistically significant. Mean values in this work are presented as (mean ± standard deviation).

## 3. Results

### 3.1. Pilot Study

By varying the enzyme concentration added to 7% beechwood in a batch reaction, four XOS fractions (XF) were obtained. The reaction parameters and composition of the XF are reported in Table 4. 

As shown in Table 4, the relative amount of XOS with a degree of polymerization lower or equal to six (X1 to X6 in Table 4) in the hydrolyzed xylan powder increased with increasing amounts of the enzyme Cellic CTec2. However, the absolute yield of free xylose is also notably higher for XF139. GPC analysis indicated that next to these XOS, the hydrolyzed xylan powder also contained fragments with number averaged molecular weight (Mn) ranging from 0.16 kDa (720 U/g xylan) to 12.3 kDa (0.5 U/g xylan).

The obtained XF were evaluated using anaerobic in vitro fermentation by caecal bacteria, followed by GC analysis of the produced SCFA and MCFA. Results for the SCFA and MCFA profiles are illustrated in Figure 2. Absolute SCFA and MCFA production, calculated as the sum of all measured FA, were determined for XF139 (326.17 ± 43.65 µmol/g caecal content), XF050 (227.94 ± 10.32), XF005 (280.32 ± 38.42), XF001 (220.04 ± 62.45), Blank T24 (75.39 ± 0.35) and Blank T0 (34.24 ± 11.07). Connecting letters reports comparing the means for all SCFA and MCFA by pairwise Student’s *t*-tests are provided in Table A2 (Appendix B). Absolute SCFA and MCFA production in all fermentation mixtures containing XOS was significantly higher (*p* ≤ 0.05) than for blank T24, mainly due to the increased production of acetic and butyric acid. With regard to the XFs, absolute SCFA production was highest for XF139. Production of butyric acid, an available energy source for birds, was significantly higher for XF139 compared to all other samples. Based on these results and using an EMR in semi-continuous mode, an XOS fraction with a tailored MS profile (XFCF) was prepared for the in vivo experiment. 

### 3.2. In Vivo Animal Experiment

#### 3.2.1. Characterization of XFCF and the Experimental Diets

To assess the prebiotic properties of the produced XOS fraction, an in vivo broiler feed experiment was designed. An XOS fraction XFCF was prepared in an EMR run in a semi-continuous mode. In this reactor, the enzymatic conversion was coupled to filtration over a hollow fiber membrane module (10 kDa molecular weight cut-off). The reaction parameters and composition of the resulting xylan fractions are reported in Table 5. 

The lyophilized powder obtained from the permeate of the EMR contained a concentration of 153.7 g/kg of X1 to X6 XOS. This powder mainly contained X3 and X4 XOS, while the concentration of X2, X5, and X6 XOS was somewhat lower. Much lower concentrations of xylose were detected, which is desirable for animal feed. Next to these X1 to X6 XOS, bigger fragments with an average Mn of 6 kDa were detected in the hydrolyzed xylan powder. 

In order to obtain more insight into the chemical characteristics of the experimental diets, the following parameters were determined in the pure XFCF preparation, and the three experimental diets FCON (no XFCF supplemented), FXFCF0.2 (0.2% *wt/wt* XFCF), and FXFCF0.5 (0.5% *wt/wt* XFCF): DM, total MS, WE MS, free MS, reducing end xylose, TOT-AX, WE-AX, and avDP.

The total and water-extractable MS profiles of XFCF and the experimental diets (FCON, FXFCF0.2, FXFCF0.5), used for the calculation of TOT-AX and WE-AX content of the samples, are reported in Table 6. 

As Table 6 shows, almost no arabinose (0.67%) was detected in the total MS profile of the pure XFCF fraction, indicating that the produced XCFC fraction was not contaminated by arabinoxylo-oligosaccharides. The purity of the XOS in XFCF could be calculated as 0.88 × [xylose %] available in this fraction. Due to interference linked to the complexity of the feed matrix and the high amount of arabinoxylan in the feed compounds maize and wheat, the XOS added to the feed could not be detected in the total MS profile. However, since XFCF has a water-extractability of 98%, calculated as the ratio of WE-xylose over TOT-xylose, and the WE-AX fraction of the control feed (FCON) only makes up 4% of the TOT AX fraction, the WE MS profile was analyzed to detect the XOS enrichment in the experimental diets. 

Similarly, almost no WE arabinose (0.61%) could be detected in the pure XFCF fraction, confirming the sample was not contaminated by arabinoxylo-oligosaccharides. The XOS purity of the XFCF fraction was estimated at 63.67%. A difference in total XOS levels between the control feed FCON (0.12 ± 0.01% *wt/wt*) and the experimental diets FXFCF0.2 (0.26 ± 0.00%) and FXFCF0.5 (0.45 ± 0.01%) was observed (Table 6). Based on the XCFC incorporation percentages (0.02% and 0.05%) corrected for the XOS purity of the XCFC fraction, the expected XOS levels in the experimental feed preparations could be confirmed perfectly. Results for the DM, free MS, reducing end xylose, and avDP are reported in Table 7.

The avDP of xylose chains is decreasing as incorporation levels of XFCF increase, confirming the incorporation of the XFCF feed supplement in the experimental diets at different levels. The estimated free xylose content of the XFCF is 1.23% *wt/wt*, or 12.28 g/kg, which is higher than the concentration value obtained immediately after the xylan fermentation (6.00 g/kg; Table 5).

#### 3.2.2. In Vivo Experiment on Chickens

To evaluate the effect of XOS supplementation on broiler performance, body weight (BW) and feed intake (FI) were measured. Subsequently, an estimation for feed conversion ratio (FCR) was made for both groups: the control group (1.46 ± 0.01, *n* = 2) and the group that received the XOS-supplemented feed (1.51 ± 0.04, *n* = 2). The evolution of BW for both groups is displayed in Figure 3. Both for FCR and BW, no statistically significant differences (*p* > 0.05) between the groups could be detected. The mortality rate of the birds was not affected by dietary treatments and was lower than 4.2% in all groups.

The net weights of the duodenum, jejunum, ileum, and caeca were registered, and the sum of the four intestinal sections was used to calculate the net gastro-intestinal tract (GIT) weight and the GIT/BW ratio. The results are reported in Figure 4. No statistically significant differences (*p* > 0.05) were found for any parameter on any measurement day.

#### 3.2.3. Microbiological Analysis of Caecal Content

During the animal experiment, animals were dissected on d8, d11, and d15, and the content of both caeca was collected and pooled. Microbiological analysis of the caecal content was performed using selective agar plates. Three groups of caecal bacteria were enumerated: *Enterobacteriaceae*, *Bifidobacteria*, and aerobic and aerotolerant anaerobic *lactobacilli*. The results are displayed in Figure 5.

The amount of caecal aerobic and aerotolerant anaerobic *lactobacilli* was significantly higher on d8 compared to d11 and d15 for both the XOS and the CON group. For *Bifidobacteria*, a similar evolution was observed for the CON group but not for the XOS group. For the latter, no significant differences were found between the different sampling days. On d11 and d15, the concentration of *Bifidobacteria* in the CON group was significantly lower compared to the XOS group. *Enterobacteriaceae* counts did increase for both groups during the experimental period. However, a sharper increase was noticed for broilers who received the XOS-supplemented diet compared to their CON counterparts. 

#### 3.2.4. SCFA and MCFA Analysis 

The volatile FA profile (acetic, propionic, butyric, isobutyric, valeric, isovaleric, caproic, and isocaproic acid) of the caecal content was determined by DEE of the samples, followed by GC. The results are reported in Figure 6. 

The average acetic acid concentration in the caecal samples did not differ among dietary treatment and age at sampling. The concentration of propionic- and butyric acid increased significantly in the caecal content of CON birds between d8 and d15. For XOS birds, on the other hand, the propionic acid concentration decreased significantly during this period. The propionic acid concentration was significantly higher for XOS birds compared to CON birds on d8 but not from the next sampling day (d11) onwards. The valeric acid concentration was significantly higher in CON birds as compared to XOS birds on d15. Isobutyric- and isovaleric acid concentrations on d8 were significantly higher for XOS birds compared to CON birds. No statistical differences were found in total produced FA, calculated as the sum of all quantified FA across both groups and sampling ages. 

Caecal *Enterobacteriaceae* were found to be positively correlated (*p* < 0.05) with caecal caproic acid concentrations (R = 0.6052). Caproic acid concentrations in CON birds increased significantly between d8 and d15 and between d11 and d15. A negative correlation (*p* < 0.05) was found between *Bifidobacteria* and propionic- (R = −0.5073), isobutyric- (R = −0.5064), and valeric acid (R = −0.5674). 

## 4. Discussion

XOS are hydrolysis products of xylan, the second most abundant carbohydrate material in lignocellulosic biomass. As a result of the increasing wood consumption worldwide, the production of waste wood, for example, in the form of end-of-life wood-based products, is also increasing, providing an abundant source of cheap raw material [30]. Transforming locally available wood residues into value-added products with applications for the agro-industry is an interesting approach to valorizing waste wood. For this purpose, innovative conversion technologies are currently explored, and validation of the resulting products is needed. The objective of this exploratory research study was to investigate the potential of a tailored XOS fraction prepared by enzymatic hydrolysis of beechwood xylan as a prebiotic feed supplement for broilers. In the first part of this study, a hydrolysis study at the batch level was conducted to investigate the optimal XOS fraction profile for in vitro fermentation of caecal content of chickens as a pilot study for the in vivo experiment. Four XOS fractions were prepared by adding various enzyme amounts to the xylan solution: XF139, XF050, XF005, and XF001. The obtained XFs were evaluated using in vitro fermentation by caecal bacteria, followed by gas chromatography analysis of the produced SCFA and MCFA.

Both absolute and relative amounts of xylose polymers per fermentation volume with a degree of polymerization lower or equal to six increased with augmenting enzyme mix Cellic CTec2 concentration. As a result, the absolute yield of free xylose was also notably higher for XF139. Free xylose, competing with glucose molecules for the same intestinal transporter systems in the duodenum and jejunum, reaches the ileo-caeco-colic section of the gastro-intestinal tract only in limited quantities and, consequently, exhibits almost no prebiotic properties [40]. Moreover, too much free xylose in the gastro-intestinal system also has a negative effect, causing higher osmotic stress, which results in more water intake and very liquid feces [41]. The absolute amount of xylose polymers per fermentation volume with a degree of polymerization between two and six was highest for XF050, followed by XF139. 

Viscosity was higher for the XF that were prepared with lower enzyme concentrations. Fermentation of XF050 and XF139 resulted in a significantly higher caproic acid production as compared to XF001 and XF005 (Figure 2, Table A2). MCFA, such as caproic acid, inhibits the overgrowth of pathogenic bacteria due to their antibacterial activity, resulting in a relative increase in beneficial bacteria, improved nutrient absorption, and, consequently, better growth performance of the birds [24]. Total SCFA production was highest for XF139 (Figure 2 and Table A2), mainly due to the contribution of acetic- and butyric acid. Because an increase in SCFA concentration in the intestinal environment is often associated with an increase in beneficial bacteria due to their antibacterial and bacteriostatic properties against pathogenic bacteria, higher SCFA production is beneficial [24]. In addition, the production of SCFA butyric acid, an available energy source for the host, was significantly higher for XF139 as compared to all other XFs. This might indicate that higher inclusion of the enzyme mix yielded more hydrolysis products, which, in turn, could be fermented by probiotic bacterial strains such as *Bifidobacterium* spp. and *Lactobacilli* spp. However, it should be noted that the higher total SCFA production in XF139 might be mainly caused by the presence of a significantly higher free xylose concentration. Free xylose, although reaching the ileo-caeco-colic section of the gastro-intestinal tract only in limited quantities in an in vivo context, can serve as a substrate for fermentation by the caecal bacteria during in vitro fermentation, resulting in a higher SCFA production. Taking these factors into account and in order to fully explore the potential of an XF obtained by enzymatic hydrolysis of BX as a prebiotic feed additive, a semi-continuous EMR system was used, allowing for the production of tailored XOS in larger quantities, and an in vivo animal experiment was designed and conducted.

To ensure feed quality, XOS profiles in the experimental diets were analyzed prior to the animal experiment. Although it has been previously demonstrated that freeze drying of XOS samples and storage at low temperatures does not result in degradation due to loss of xylose linkages by hydrolysis [42,43], analysis of XFCF in this study before and after lyophilization showed that free xylose concentration, measured with two different techniques, doubled from 6.00 g/kg (HPAEC-PAD, Table 5) to 12.28 g/kg (GC, Table 7) during this process. This is clearly an issue that requires further exploration to reveal the underlying causes in order to ensure the long-term stability of XFCF and similar products. On the other hand, carbohydrases present in the feed did not seem to influence XOS availability in the feed. Indeed, taking into account the purity of XFCF, the supplemented XOS could be entirely detected in the WE fraction of the feed. 

During the animal experiment, growth performance, FCR, caecal SCFA and MCFA concentration, and microbiological analysis of the caecal content of broiler chickens were assessed. Administration of the XOS-supplemented feed did not reveal influences on the zootechnical parameters of the birds, such as growth performance, estimated FCR (*n* = 2), and growth of GIT. These results do not differ significantly from previous studies: Courtin and colleagues reported that XOS supplementation did not influence BW at d14 [23]. However, at d7, they observed a lower BW for chickens receiving an XOS-supplemented diet. In a similar study, comparable FCR values for Ross 308 chickens (d14) were reported: 1.54 for birds from the control group (versus 1.46 ± 0.01 (d15) in the present study), and 1.51 (0.1% XOS) and 1.59 (0.025% XOS) for birds that received a supplemented diet (versus 1.51 ± 0.04 in the present study). Both in the present study and the study performed by Craig and colleagues, FCR was not statistically different for both groups [19]. In both studies, birds did not achieve the performance objectives of Aviagen for male broilers on d15 (FCR 1.14) [44]. This could be due to a number of reasons, including feed composition. One possible explanation could be the low protein levels in the feed (18% CP in the present study, 20% in the study by Craig et al.), which is a known cause of reduced growth performance of broilers [19,45]. 

The intestinal microbiome plays a pivotal role in maintaining the overall intestinal health of the host, regulating various physiological functions such as the integrity of the epithelial barrier, digestion, nutrient absorption, inflammatory balance, and mucosal immunity. These functions are intricately modulated via extensive interactions with the host, its environment, the chyme, and interactions among individual intestinal bacteria [46]. In a state of health, the intestinal microbiome achieves eubiosis, increasing the resistance of the gastro-intestinal tract against pathogenic colonization via mechanisms like competitive exclusion [47]. Dysbiosis, on the other hand, is a known trigger of inflammation, which, in turn, triggers dysbiosis, causing a vicious cycle to emerge: a shift in microbiota leads to inflammation and oxidative stress, leading to a dysfunctional mucosal layer. This leads to serious morphological and functional alterations in the gastro-intestinal tract. A gastro-intestinal tract with poor functionality leads to poor digestion of feed and nutrients, which creates an oversupply of nutrients in the lumen. Specifically, excess protein can trigger intestinal inflammation. When not efficiently digested and absorbed, dietary proteins become a substrate for microbiota, favoring some bacterial groups while disfavoring others, causing a disbalance and further increasing the shift in microbiota [48]. Given this strong bidirectional relationship between intestinal health, nutrition, and intestinal microbiota composition, microbiological analysis of the caecal content of the broiler chicks was performed, and three groups of bacteria were enumerated: *Enterobacteriaceae*, *Bifidobacteria*, and aerobic and aerotolerant anaerobic *lactobacilli*. Count of caecal *Enterobacteriaceae*, a family that comprises many known pathogenic bacterial strains such as *Salmonella*, *Shigella*, and avian pathogenic *Escherichia coli*, was found to be positively correlated with caecal caproic acid concentrations. These findings seem to contradict previous research from Kumar and colleagues, who reported that MCFA inhibits the overgrowth of pathogenic bacteria because of their potent antibacterial activity, resulting in an increase in beneficial bacteria [24]. Nevertheless, results should be interpreted with caution, given the complexity of the intestinal microbial ecosystem. Caproic acid concentrations in CON birds increased significantly between d8 and d11 but not in XOS birds. Unlike for XOS birds, caecal *Enterobacteriaceae* counts in CON-fed broilers were not significantly different between d8 and d15, suggesting that this increase for the latter group might have been slower. However, it should be noted that pairwise comparison per day does not reveal significant differences between both groups. A similar trend could be observed for butyric acid. No significant differences between the two groups were found, but butyric acid increased significantly for CON broilers between d8 and d15, whereas for XOS birds, butyric acid levels remained at the same level throughout the entire duration of this experiment. These results are in line with findings from a similar study, in which xylanase was supplemented to the experimental diet to generate XOS in vivo in the GIT, and no significant differences in caecal butyric acid proportions between broilers fed the control diet and those fed xylanase-supplemented diets were found either [45]. Butyrate, an SCFA produced by probiotic bacteria and used as an energy source by the colon epithelial cells, can exert anti-inflammatory effects by inhibiting the NF-κB pathway. Additionally, in lipopolysaccharide (LPS)-activated macrophages, butyrate suppresses the expression of pro-inflammatory cytokines such as IFN-γ, IL-1β, and IL-6 [49].

Furthermore, our results provide added support for the view that hydrolysis products of beechwood xylan stimulate bifidobacterial growth since, on d11 and d15 *Bifidobacterium*, counts were significantly higher in the XOS group compared to the CON group [23,33]. *Bifidobacteria* are considered probiotic microorganisms and play a critical role in maintaining immune homeostasis in chickens. Probiotic bacteria, via the upregulation of MUC2 in goblet cells, have the capability to enhance intestinal barrier integrity by increasing the production of mucus. Additionally, these bacteria contribute to the reduction in tight junction permeability by upregulating zonulin [50]. Certain probiotic strains, including *Bifidobacteria*, can synthesize vitamins and exhibit antioxidant properties themselves by scavenging free radicals [18,51]. Previous research indicated that certain probiotic strains of *Bifidobacterium* spp. grow more efficiently on xylobiose, xylotriose, or XOS than on xylose because they possess a specific uptake mechanism for XOS and intracellularly located carbohydrases, providing them with a competitive advantage over other strains that can only take up monosaccharides [25]. Prebiotic carbohydrate fermentation by specific strains, such as *Bifidobacterium* spp., has been linked to a reduced protein fermentation, which is beneficial to the host health since bacterial protease pathways, including deamination of amino acids and decarboxylation, can produce potentially toxic compounds with carcinogenic properties [22,52]. In this study, the evaluation of longer-term effects on increased *Bifidobacteria* was not conducted. It is important to note that potential positive impacts on zootechnical parameters may not manifest immediately. Nevertheless, investigating these effects over an extended duration could provide valuable insights, including considerations of additional parameters such as antibiotic use, which might exhibit reductions over time.

## 5. Conclusions

In conclusion, XOS with a tailored MS profile prepared using enzymatic hydrolysis of beechwood xylan has the potential to serve as a prebiotic feed supplement for broilers. Results from the beechwood hydrolysis in batch set-up indicated that higher enzyme concentrations in the hydrolysis process yield a product with lower viscosity and a higher total SCFA-, MCFA-, and butyric acid production during in vitro anaerobic fermentation by caecal bacteria. Supplementation of the tailored XFCF to the broiler diet resulted in higher *Bifidobacterium* counts, beneficial to the health of the birds, on d11 and d15 compared to the control group. However, administration of an XOS-supplemented feed did not reveal influences on the zootechnical parameters of the birds, such as growth performance, FCR, and growth of GIT. Whether supplementation of XOS results in beneficial effects in the long term remains to be clarified in further in vivo work.

## Figures and Tables

**Figure 1 animals-13-03511-f001:**
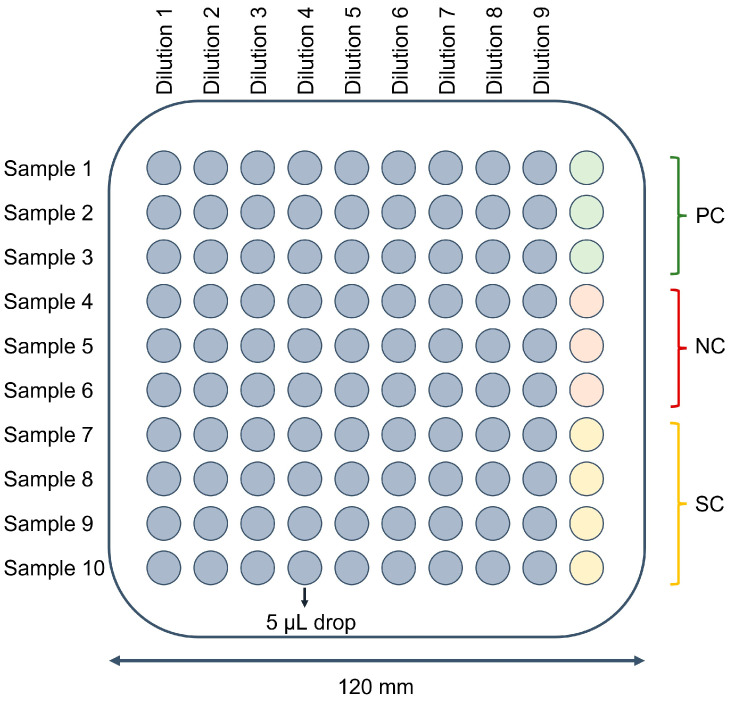
Schematic overview of the organization of a selective agar plate for microbiological analysis based on the drop plate method by Chen and colleagues [29]. Per plate, ten samples could be tested, each in nine dilutions, and three positive bacterial controls (PC), three negative bacterial controls (NC), and four peptone physiological salt solution sterility controls (SC) were included. Dilutions of the samples ranged from 500 to 500 × 10^−8^ mg caecal content per mL.

**Figure 2 animals-13-03511-f002:**
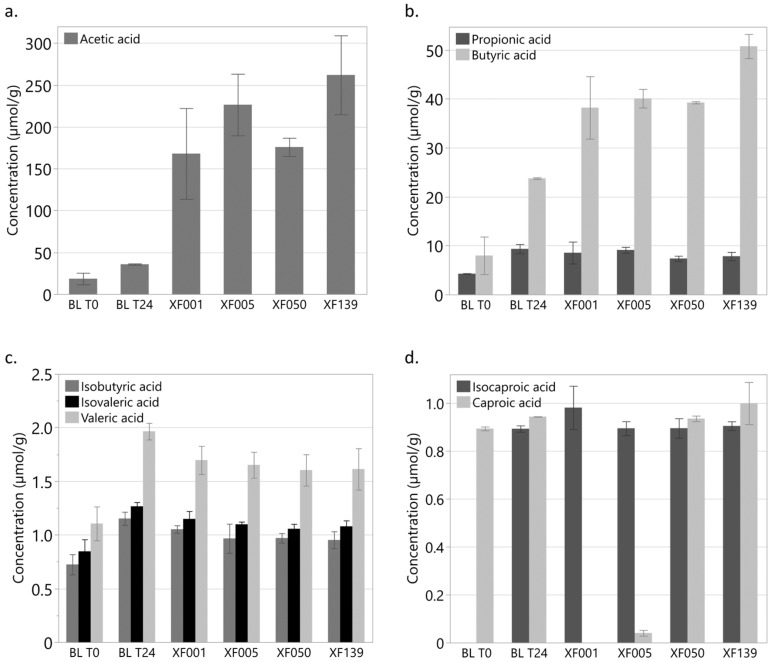
Volatile fatty acid (FA) profile of the caecal content of chickens after 24 h of in vitro fermentation for the four xylo-oligosaccharide (XOS) fractions, prepared by adding various enzyme volumes to a 7.0% *w/v* xylan solution: XF139 (13.9% *vol/wt*, calculated as Cellic CTec2 (mL) per xylan (g)), XF050 (0.50% *vol/wt*), XF005 (0.05% *vol/wt*), XF001 (0.01% *vol/wt*), blank T24 (BL T24; 0.00% *vol/wt*), and BL T0 (0.00% *vol/wt*). (**a**) Average acetic acid production (µmol/g caecal content); (**b**) Average propionic- and butyric acid concentration (µmol/g caecal content); (**c**) Average isobutyric-, isovaleric- and valeric acid concentration (µmol/g caecal content); (**d**) Average isocaproic and caproic acid concentration (µmol/g caecal content). All preparations except blank T0 were incubated for 24 h at 41 °C under anaerobic conditions after the addition of a solution containing caecal bacteria from chickens. Each error bar represents (mean ± standard deviation). Connecting letters reports comparing the means for all SCFA and MCFA by pairwise Student’s *t*-tests are provided in Table A2 (Appendix B).

**Figure 3 animals-13-03511-f003:**
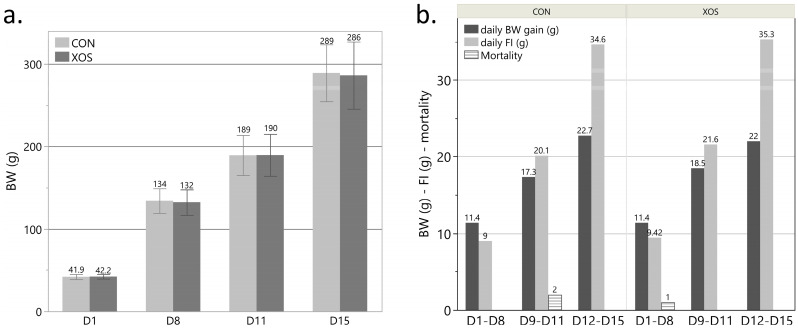
(**a**) Average body weight for the control group (CON) and the group receiving the xylo-oligosaccharide supplemented feed (XOS) on day 1 (d1; *n* = 45), d8 (*n* = 45), d11 (*n* = 30) and d15 (*n* = 15). Each error bar represents (mean ± standard deviation). No statistically significant differences (*p* > 0.05) were found between the groups for all days. (**b**) Daily body weight (BW) gain, daily feed intake (FI), and absolute mortality for the control group (CON) and the group receiving the XOS-supplemented feed during the periods d1–d8 (*n* = 45), d9–d11 (*n* = 30) and d12–d15 (*n* = 15). No statistically significant differences (*p* > 0.05) were found between the groups for all days.

**Figure 4 animals-13-03511-f004:**
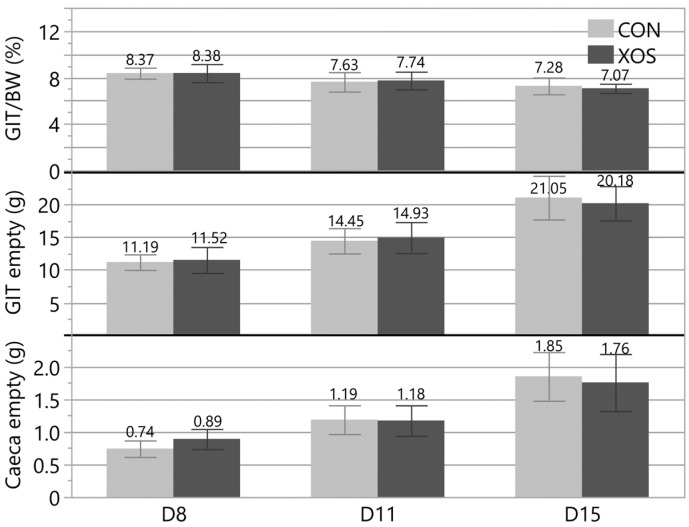
Average gastro-intestinal tract (duodenum, jejunum, ileum and caeca) over body weight ratio (GIT/BW), net GIT weight, and net caeca weight for the control group (CON) and the group receiving the xylo-oligosaccharide supplemented feed (XOS) on day 8 (D8; *n* = 15), d11 (*n* = 15) and d15 (*n* = 15). Each error bar represents (mean ± standard deviation). No statistically significant differences (*p* > 0.05) were found between the groups for all parameters and all days.

**Figure 5 animals-13-03511-f005:**
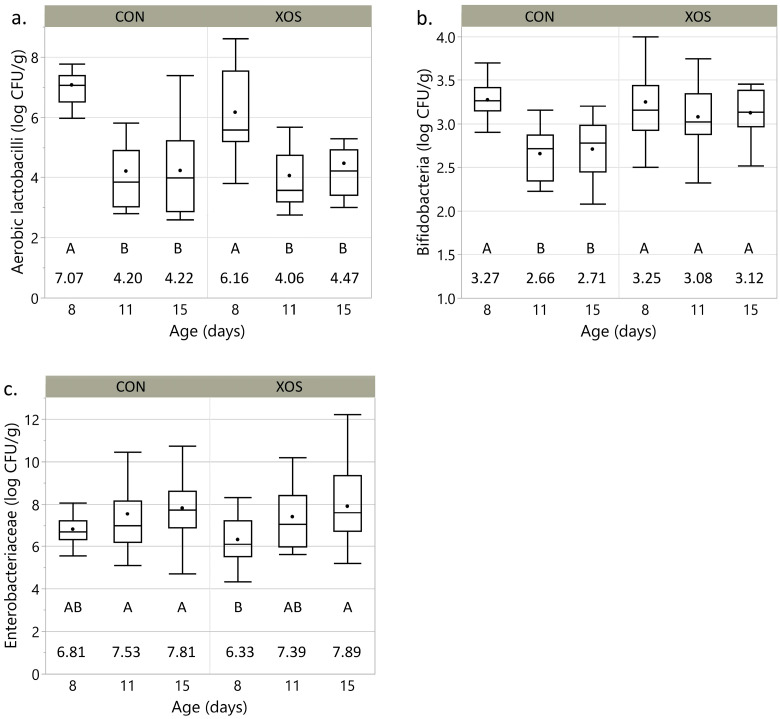
Average logarithmic number of colony-forming units (log CFU/g) of three groups of caecal bacteria [(**a**) aerobic *lactobacilli*; (**b**) *Bifidobacteria*; (**c**) and *Enterobacteriaceae*] in caecal content of chickens receiving a control diet (CON) versus a xylo-oligosaccharide supplemented diet (XOS) at the age of 8, 11 and 15 days. Comparison of the means for all groups was performed by pairwise Student’s *t*-tests and represented using a connecting letter report. Levels not connected by the same letter are significantly different (*p* ≤ 0.05). Mean values (log CFU/g) are reported under the connecting letter report.

**Figure 6 animals-13-03511-f006:**
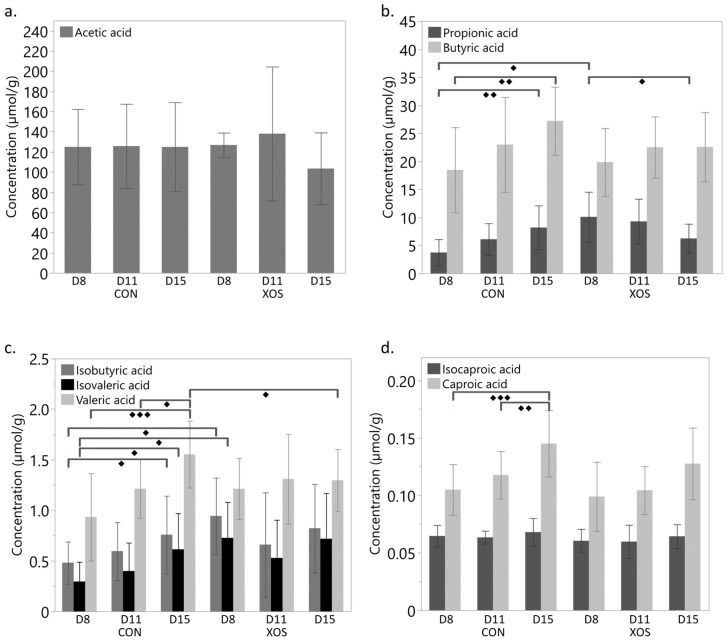
Volatile FA profile of the caecal content of chickens receiving a control diet (CON) versus a xylo-oligosaccharide supplemented diet (XOS) at the age of 8 (d8), 11 and 15 days. No statistical differences were found in total produced FA, calculated as the sum of all quantified FA across both groups and sampling ages. (**a**) Average caecal acetic acid concentration (µmol/g caecal content); (**b**) Average caecal propionic- and butyric acid concentration (µmol/g caecal content); (**c**) Average isobutyric-, isovaleric- and valeric acid concentration (µmol/g caecal content); (**d**) Average isocaproic- and caproic acid concentration (µmol/g caecal content). Statistical differences between groups were pairwise tested per treatment and per day. Significant differences are indicated by ◆ (*p* ≤ 0.05), ◆◆ (*p* ≤ 0.01), and ◆◆◆ (*p* ≤ 0.001). Each error bar represents (mean ± standard deviation).

**Table 1 animals-13-03511-t001:** Fermentation mixture composition and incubation conditions for the in vitro fermentation of the xylo-oligosaccharide fraction (XF) preparations.

Group	Cellic CTec2 Loading (U/g Xylan)	Incubation Conditions			Composition Fermentation Mixture		
		T (°C)	O_2_	t (h)	XF Fraction (mL)	Sterile PBS (mL)	42.5% *vol/vol* Caecal Content (mL)
XF139 ^a^	719	41	AN ^b^	24	0.6	2.4	1
XF050	25	41	AN	24	0.6	2.4	1
XF005	2.6	41	AN	24	0.6	2.4	1
XF001	0.5	41	AN	24	0.6	2.4	1
B-T24	-	41	AN	24	0	3	1
B-T0	-	NA	NA	0	0	3	1

^a^ Abbreviations: XF = xylo-oligosaccharide fraction; T = temperature in degrees Celcius; B-T24 = blank T24, caecal content solution incubated during 24 h; B-T0 = blank T0, caecal content solution that did not undergo incubation; O_2_ = presence of oxygen; AN = anaerobic conditions; t = time of incubation in h; NA = not applicable; PBS = phosphate-buffered saline solution. ^b^ Anaerobic conditions were obtained in an anaerobic jar using the Anaerocult A system (REF 1.32381.0001, Merck, Darmstadt, Germany). The absence of oxygen was confirmed using an Anaerotest Strip (REF 1. 32371.0001, Merck) that was placed in the anaerobic jar and regularly checked.

**Table 2 animals-13-03511-t002:** Experimental design of the animal study.

		Number of Animals Sampled		
Group	Treatment	d8 ^a^	d11	d15
CON (CA + CB)	Control feed	15	15	15
XOS (XA + XB)	Feed supplemented with XOS:	15	15	15
	d1–d8: 0.20% XFCF supplementation = 0.13% XOS supplementation			
	d9–d15: 0.50% XFCF supplementation = 0.32% XOS supplementation			
Total		30	30	30

^a^ Abbreviations: d = age of the flock in days; XOS = xylo-oligosaccharides; XFCF = tailored hydrolyzed xylan fraction intended as a chicken feed supplement.

**Table 3 animals-13-03511-t003:** Selective agar plate composition and incubation conditions for the enumeration of bacterial groups *Enterobacteriaceae*, *Bifidobacteria*, and aerobic and aerotolerant anaerobic *lactobacilli*.

Group	Composition Selective Agar Plates	Incubation Conditions ^a^			Bacterial Strains for Quality Control Selective Agar Plates ^c^	
		T (°C)	O_2_	t (h)	Positive Control	Negative Control
*Enterobacteriaceae*	Violet Red Bile Glucose agar (REF CM0485B, Thermo Fisher Scientific)	37	AE	15	Avian Pathogenic *Eschericia coli*	*Enterococcus cecorum*
*Bifidobacteria*	Wilkins-Chalgren Anaerobe agar (REF CM0619B, Thermo Fisher Scientific), supplemented with 1.00 mL/L glacial acetic acid (REF A6283, Merck) and 0.1 g/L mupirocin ^d^ (REF A4718.0005, VWR, Radnor, PA, USA)	37	AN ^b^	45	*Salmonella enterica*	*Clostridium perfringens*, *Lactobacillus salivarius*
Aerobic and aerotolerant anaerobic *lactobacilli*	Rogosa agar (REF CM0627B, Thermo Fisher Scientific), supplemented with 1.32 mL/L glacial acetic acid (REF A6283, Merck)	37	AE	90	*Lactobacillus salivarius*	*Staphylococcus aureus*

^a^ Abbreviations: T = temperature in degrees Celcius; O_2_ = presence of oxygen; AE = aerobic conditions; AN = anaerobic conditions; t = time of incubation in h. ^b^ Anaerobic conditions were obtained in an anaerobic jar using the Anaerocult A system (REF 1.32381.0001, Merck, Darmstadt, Germany). The absence of oxygen was confirmed using an Anaerotest Strip (REF 1. 32371.0001, Merck) that was placed in the anaerobic jar and regularly checked. ^c^ During the experiment, a positive and negative control were included on every selective agar plate to confirm the selective quality of the plate. ^d^ Differentiation between bacteria from the genera *Bifidobacteria* and *lactobacilli* is often challenging because their cultural and biochemical properties overlap. However, *Bifidobacterium* spp. are, in contrast to *Lactobacillus* spp., resistant to growth inhibition by mupirocin. This antibiotic agent also inhibits the growth of several other genera of Gram-positive bacteria, including other closely related lactic acid bacteria commonly found in caeca of broilers [21,39].

**Table 4 animals-13-03511-t004:** Reaction parameters and characterization of the hydrolyzed xylan fractions obtained during enzymatic hydrolysis of 7% beechwood xylan in a batch set-up.

Hydrolyzed Xylan Fraction	Reaction Parameters ^a^			Composition XOS Fraction (g/kg) ^b^							Molecular Weight (kDa)	
	Cellic CTec2 Loading (U/g Xylan)	t (h)	pH	X1	X2	X3	X4	X5	X6	>X6	Mw	Mn
XF139	719	23.2	5.05	35.94	4.14	0.55	1.16	0.54	0.13	957.54	3.1	0.16
XF050	25	24.1	-	4.99	8.86	3.29	0.75	0.42	0.45	981.24	31.5	1.10
XF005	2.6	18.3	6.28	0.29	1.14	1.30	1.09	1.01	0.84	994.33	633.6	3.80
XF001	0.5	24.0	6.94	0.09	0.34	0.39	0.40	0.49	0.37	997.92	1443.0	12.30

^a^ Abbreviations: XOS = xylo-oligosaccharides; XF = hydrolyzed xylan fraction; t = hydrolysis time; Mw = weight averaged molecular weight; Mn = number averaged molecular weight. ^b^ Composition (g/kg) = g component per kg xylan fragments XOS; X1 = xylose; X2 = xylobiose; X3 = xylotriose; X4 = xylotetraose; X5 = xylopentaose; X6 = xylohexaose.

**Table 5 animals-13-03511-t005:** Reaction parameters and characterization of hydrolyzed xylan fraction prepared in a semi-continuous enzyme membrane reactor for the in vivo animal experiment.

	Reaction Parameters ^a^			Composition Hydrolyzed Xylan (g/kg) ^b^						Molecular Weight (kDa)	
Hydrolyzed Xylan Fraction	Cellic CTec 2 Loading (U/g Xylan)	t (h)	pH	X1	X2	X3	X4	X5	X6	Mw	Mn
XFCF	5.4	19.0	5.05	6.00	22.55	52.05	31.75	24.00	17.35	7.2	6.0

^a^ Abbreviations: XFCF = tailored hydrolyzed xylan fraction intended as chicken feed supplement; t = hydrolysis time; Mw = weight averaged molecular weight; Mn = number averaged molecular weight. ^b^ Composition (g/kg) = g component per kg xylan fragments XOS; X1 = xylose; X2 = xylobiose; X3 = xylotriose; X4 = xylotetraose; X5 = xylopentaose; X6 = xylohexaose.

**Table 6 animals-13-03511-t006:** Total and water-extractable monosaccharide profile of the prepared hydrolyzed xylan fraction and calculation of the total amount of arabinoxylan and water-extractable arabinoxylan.

Sample ^a^	Total Monosaccharide Profile ^c^ (% *wt/wt*)					Calculations ^b^	
	**L-arabinose**	**D-xylose**	**D-mannose**	**D-galactose**	**D-glucose**	**TOT-AX%**	**TOT-A/X**
FCON	3.12 ± 0.12	3.40 ± 0.25	0.80 ± 0.05	2.41 ± 0.27	51.74 ± 2.23	5.73 ± 0.24	0.92 ± 0.08
FXFCF0.2	3.12 ± 0.13	3.71 ± 0.06	0.81 ± 0.10	2.79 ± 0.19	48.50 ± 1.48	6.01 ± 0.12	0.84 ± 0.04
FXFCF0.5	2.88 ± 0.08	3.66 ± 0.30	0.78 ± 0.06	2.57 ± 0.13	50.61 ± 0.99	5.76 ± 0.27	0.79 ± 0.07
XFCF	0.67 ± 0.00	73.97 ± 1.14	0.17 ± 0.01	0.84 ± 0.02	3.59 ± 0.09	65.68 ± 1.00	0.01 ± 0.00
	**Water-Extractable Monosaccharide Profile (% *wt/wt*)**					**Calculations**	
	**L-arabinose**	**D-xylose**	**D-mannose**	**D-galactose**	**D-glucose**	**WE-AX%**	**WE-A/X**
FCON	0.15 ± 0.00	0.12 ± 0.01	0.50 ± 0.06	0.97 ± 0.01	3.10 ± 0.06	0.24 ± 0.01	1.24 ± 0.09
FXFCF0.2	0.17 ± 0.00	0.26 ± 0.00	0.58 ± 0.06	1.06 ± 0.01	3.41 ± 0.09	0.38 ± 0.00	0.65 ± 0.00
FXFCF0.5	0.17 ± 0.00	0.45 ± 0.01	0.55 ± 0.03	1.01 ± 0.01	3.11 ± 0.09	0.54 ± 0.01	0.37 ± 0.01
XFCF	0.61 ± 0.02	72.35 ± 1.16	0.15 ± 0.00	0.79 ± 0.06	3.30 ± 0.15	64.2 ± 1.02	0.01 ± 0.00

^a^ Sample composition: XFCF = tailored hydrolyzed xylan fraction intended as chicken feed supplement; control feed FCON = no XFCF supplemented; experimental feed FXFCF0.2 = 0.2% *wt/wt* XFCF supplemented; and experimental feed FXFCF0.5 = 0.5% *wt/wt* XFCF supplemented. ^b^ Calculations: total amount of arabinoxylan (TOT-AX (%)) was calculated as 0.88 × [(L-arabinose %) + (xylose %)]. Similarly, the total amount of water-extractable arabinoxylan (WE-AX (%)) was calculated as 0.88 × [(WE-L-arabinose %) + (WE- xylose %)]. A conversion factor of 0.88 was used to account for the release of water molecules by monosaccharides during the polymerization reaction. Total and WE arabinose to xylose (A/X) ratios (TOT-AX A/X ratio, WE-AX A/X ratio) were calculated by dividing the arabinose by the xylose content of the samples and their aqueous extracts, respectively. ^c^ In order to assess the total and water-extractable monosaccharide profiles of the feed samples, L-arabinose, D-galactose, D-glucose, D-mannose, and D-xylose concentrations were determined.

**Table 7 animals-13-03511-t007:** Moisture content, reducing end and free monosaccharide content, and average degree of polymerization of the xylo-oligosaccharide fraction and the experimental diets.

Sample ^a^	Reducing End Sugar	Free Monosaccharides		AvDP Xylose Chain ^b^	Moisture Content (%)
	Xylose (%)	Arabinose (%)	Xylose (%)		
FCON	0.00 ± 0.00	0.00 ± 0.00	0.00 ± 0.00	38.06 ± 3.64	12.90 ± 0.05
FXFCF0.2	0.02 ± 0.00	0.01 ± 0.00	0.01 ± 0.00	11.96 ± 0.20	12.48 ± 0.06
FXFCF0.5	0.05 ± 0.00	0.01 ± 0.00	0.01 ± 0.00	9.19 ± 0.28	12.84 ± 0.09
XFCF	11.89 ± 0.52	0.09 ± 0.00	1.23 ± 0.04	6.08 ± 0.28	7.34 ± 0.03

^a^ Sample composition: XFCF = tailored hydrolyzed xylan fraction intended as chicken feed supplement; control feed FCON = no XFCF supplemented; experimental feed FXFCF0.2 = 0.2% *wt/wt* XFCF supplemented; and experimental feed FXFCF0.5 = 0.5% *wt/wt* XFCF supplemented. ^b^ The average degree of polymerization (avDP) of the xylose chain was calculated as [(L-arabinose%) + (D-xylose%)] (Table 6), divided by the amount of reducing end D-xylose (%).

## Data Availability

The data presented in this study are available on request from the corresponding author.

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
