# Peer review of "Toward Renewable-Based Prebiotics from Woody Biomass: Potential of Tailored Xylo-Oligosaccharides Obtained by Enzymatic Hydrolysis of Beechwood Xylan as a Prebiotic Feed Supplement for Young Broilers"

_animals, 2023, doi:10.3390/ani13223511_

Round 1

Reviewer 1 Report

Comments and Suggestions for Authors

The purpose of this study was to determine the potential of xylo-oligosaccharides prepared by enzymatic hydrolysis of beechwood xylan as a prebiotic feed supplement for chicken broilers. The Introduction chapter provides an overview of the world's knowledge on this subject. The material used in the research is sufficiently numerous, Materials and Methods is correct. The Results chapters are needed some corrections. The discussion is exhaustive. Summary of the results are correct. The proposed changes are listed below.

General comments:

Prepare the article in accordance with the instructions for the authors:

For first and last name use initials (same as in Author Contributions section), and an e-mail for each co-author of this article

For significance, use a low letter "p" in italic instead of "p" in the main article

In References section for page ranges use long (-) from the symbol function, instead of short (-) from the keyboard for item 12, and also an italic for journal name and volume number and year in bold for all references.

Detailed comments:

L38 (SCFA and MCFA) because the article presents data for short chain fatty acid and medium fatty acid but not their proportions

L69 In EU countries, antibiotics can currently be used therapeutically and prophylactically, but not as AGPs

L388, 415 etc. lowercase "p" in italic instead of "p"

L417 Only XF139 is the highest, please delete and XF005

Figure 2 – no significance markings in Figures 2a–c

Table 6 No description for D-mannose, D-galactose, D-glucose

L486-487 BW (body weight), FI (feed intake), and FCR (feed conversion ratio) instread of current form

No Figures for FI, FCR, mortality data over the period of 1-15 days for the control (CON) and experimental (XOS) groups, please add a new figure

L546 * (p £ 0.05), ** (p £ 0.01), and *** (p £ 0.001) instead of current form

L550 „D15 DOS” also?

L561-564 Please move to the appropriate place where the features are described

L682 SCFA and MCFA

Reviewer 2 Report

Comments and Suggestions for Authors

Despite a well-designed and presented study, there are doubts about the relevance of this manuscript to the journal's profile. Abstract, Introduction and Conclusion need to be strengthened in terms of general biology and zoology, maybe animal physiology, rather than chemistry, biochemistry, microbiology and biotechnology. In the methods section, the use of a complex enzyme preparation containing cellulases as the main activities seems strange. Either the choice of this enzyme should be justified or the high xylanase activity should be indicated.

Reviewer 3 Report

Comments and Suggestions for Authors

This research aims to find alternatives to antibiotics for promoting poultry health and maintaining production efficiency and safety. The authors explore the potential use of xylo-oligosaccharides as a prebiotic feed supplement for broilers. The results indicate that higher enzyme concentrations in production lead to increased butyric acid production during in vitro fermentation by intestinal bacteria. Furthermore, supplementation of these tailored xylo-oligosaccharides in the broiler diet results in higher Bifidobacterium population in intestinal contents.

Major comments:

1. Supplementation of XFCF increased about 60-70% of Bifidobacterium's population but without significant impact to the growth of broilers. The authors should be cautious about stating any actual beneficial effect from XFCF feeding in the conclusion.

2. Figure 6, why are there larger differences in concentrations of FA in control samples than XOS feed samples from D8 to D15? Does it mean XOS feeding make FA production be more stable?

3. Section 2.2.2, how were the control and two XOS diets prepared? Did authors add different amounts of XOS directly into FCON to make FXFCF0.2 and FXFCF0.5? If so, why wasn't the total monosaccharide profiles proportional to the total concentration of XOS?

4. Figure 5b, the figure can't show any sharper increase clearly (less than 1 log). Could author provide any result of quantitative analysis?

5. Why did authors change the concentration of feed XOS during the animal experiments?

Minor comments:

1. Figure 5, please explain the mean of "A", "B" and "AB", also label the p values in the legend.

2. Line 598-599, was it shown in figure 2 not table 5? Also why did XF050 has lower production of SCFA than XF139 and XF005?
